# Ultra-high spin emission from antiferromagnetic FeRh

Dominik Hamara[1], Mara Strungaru[2], Jamie R. Massey[3,4,5], Quentin Remy[6], Xin Chen[7], Guillermo Nava Antonio[1], Obed Alves Santos[1], Michel Hehn[8], Richard F. L. Evans[2], Roy W. Chantrell[2], Stéphane Mangin[9], Caterina Ducati[7], Christopher H. Marrows[3], Joseph Barker[3] ✉ & Chiara Ciccarelli[1] ✉

An antiferromagnet emits spin currents when time-reversal symmetry is broken. This is typically achieved by applying an external magnetic field below and above the spin-flop transition or by optical pumping. In this work we apply optical pump-THz emission spectroscopy to study picosecond spin pumping from metallic FeRh as a function of temperature. Intriguingly we find that in the low-temperature antiferromagnetic phase the laser pulse induces a large and coherent spin pumping, while not crossing into the ferromagnetic phase. With temperature and magnetic field dependent measurements combined with atomistic spin dynamics simulations we show that the antiferromagnetic spin-lattice is destabilised by the combined action of optical pumping and picosecond spin-biasing by the conduction electron population, which results in spin accumulation. We propose that the amplitude of the effect is inherent to the nature of FeRh, particularly the Rh atoms and their high spin susceptibility. We believe that the principles shown here could be used to produce more effective spin current emitters. Our results also corroborate the work of others showing that the magnetic phase transition begins on a very fast picosecond timescale, but this timescale is often hidden by measurements which are confounded by the slower domain dynamics.

FeRh has long been studied due to its first-order phase transition from a collinear antiferromagnet (AF) to a ferromagnet (FM) near room temperature[1]. There is technological interest in a material that undergoes a significant change in its structural, magnetic, electrical, and thermal properties with just a small change in temperature[2,3] or strain[4,5]. FeRh has also opened up a rich playground of fundamental studies to understand the nature of the phase transition and the parameters that trigger it. These studies have brought to light an intricate picture where the spin, electronic, and lattice degrees of freedom intertwine.

The timescale of the phase transition triggered by laser pumping has been probed using techniques such as time-resolved X-ray diffraction and absorption spectroscopies[6–9], magneto-optical Kerr effect[10,11] and double pump-THz emission spectroscopy[12]. These studies showed that the ferromagnetic domain nucleation time is in the range of tens of picoseconds. Long-range ferromagnetic order in the direction of an external magnetic field follows, with typical timescales in the range of hundreds of picoseconds, limiting the speed of the magnetic phase transition. This picture naturally leads to

[1]Department of Physics, University of Cambridge, Cambridge, UK. [2]School of Physics, Engineering and Technology, University of York, York YO10 5DD, UK. [3]School of Physics and Astronomy, University of Leeds, Leeds LS2 9JT, UK. [4]Laboratory for Mesoscopic Systems, Department of Materials, ETH Zurich, 8093 Zurich, Switzerland. [5]Paul Scherrer Institute, 5232 Villigen PSI, Switzerland. [6]Department of Physics, Freie Universität Berlin, 14195 Berlin, Germany. [7]Department of Materials Science and Metallurgy, University of Cambridge, Cambridge, UK. [8]Université de Lorraine, CNRS, IJL, F-54000 Nancy, France. [9]Institut Jean Lamour (UMR 7198), Université de Lorraine, Vandoeuvre-lès-Nancy, France. ✉e-mail: j.barker@leeds.ac.uk; cc538@cam.ac.uk

consideration of the possibility of reducing the overall timescale of the phase transition by spin-biasing the FeRh at timescales shorter than the lattice response time. In this way, long-range magnetism would be established within the lattice response time, without the need for the magnetisation of the different domains to go through a slower realignment process.

In this work, we show that the free electron bath has a strong spin-polarising action on the antiferromagnetic phase in FeRh after optical pumping. A non-zero spin polarization in the free electron bath combined with optical pumping destabilises the antiferromagnetic order of the spin-lattice and results in an enhanced generation of spin current.

Optical pump-THz emission spectroscopy is a particularly suitable technique to measure optically generated spin currents at sub-picosecond timescales. Recently, this technique has been used in several studies of FeRh-Pt heterostructures at temperatures around the AF-FM phase transition, $T_{(AF-FM)} \approx 350-370$ K[13,14]. A femtosecond optical pump is used to heat the electron bath above ambient temperature. In the FM phase, this leads to a net spin transfer to the adjacent Pt layer, where the spin current is converted into a fast current pulse via the inverse spin Hall effect (ISHE) and results in the emission of THz electro-dipole radiation. In the AF phase, the THz emission is negligible and is understood from the spin-degeneracy of

the electron bands meaning there is zero net spin transfer to the Pt. In this work, we have extended these studies over a wider temperature range of 20–420 K. We observe a previously unreported increase in the THz emission at low temperatures, consistent with a large spin current, despite the negligible magnetisation in the AF phase. This cannot be explained within a standard spintronic emitter picture, and our work provides new insight into the exchange-enhanced spin current generation in FeRh mediated by the Rh spin.

## Results

### Optical pump-THz emission from FeRh-Pt

In our measurements, we used two thin-film structures: MgO/FeRh(Pd,Ir)(30)/Pt(3.5) and MgO/FeRhPd(30), with layer thicknesses given in nanometres, respectively referred to as FeRh/Pt and FeRh samples. The composition of the FeRh/Pt sample is MgO/Fe$_{50}$Rh$_{46.8}$Pd$_{1.7}$Ir$_{1.5}$(5)/Fe$_{50}$Rh$_{47.1}$Pd$_{2.2}$Ir$_{0.7}$(10)/Fe$_{50}$Rh$_{47.2}$Pd$_{2.8}$(15)/Pt(3.5). The Pd and Ir doping gradients result in a transition temperature gradient[15,16]. The FeRh sample is a uniform Fe$_{50}$Rh$_{47.2}$Pd$_{2.8}$ film used as a control.

The layout of the experiment is shown in Fig. 1a. We applied femtosecond laser pulses incident on the sample and measured the emitted THz electric field pulses in the time domain by electro-optic sampling with a 1 mm ZnTe crystal. To indicate different directions we introduce a Cartesian coordinate system fixed with respect to our

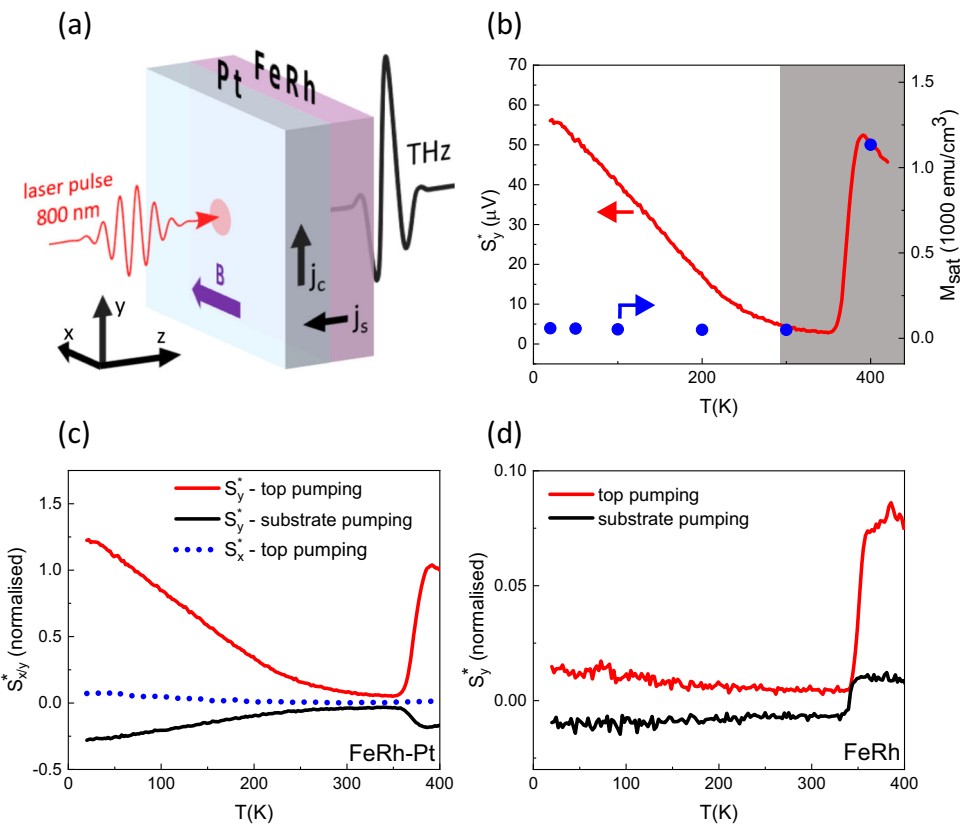

**Fig. 1 | Temperature dependence of the THz emission. a** Layout of the experiment. The pump pulse propagates in the $z$ direction, while a magnetic field can be applied in the $x$ direction. $j_s$ represents the spin current, while $j_c$ is the charge current generated in the Pt layer via the ISHE. **b** Temperature dependence of the THz emission amplitude, rescaled by the optical absorption and THz outcoupling, $S_y^*$ (red line), in top-pumping geometry with an applied magnetic field of 860 mT. On the same graph, we plot the saturation magnetisation $M_{sat}$ as a function of temperature (blue dots), derived from isothermal hysteresis loops shown in section S2 A of the SI. **c** Temperature dependence of $S_y^*$ for two different pumping geometries - top pumping (red) and substrate pumping (black) - and an applied magnetic field of 860 mT for the FeRh-Pt sample. The blue dotted line shows the

temperature dependence of the rescaled THz emission amplitude polarised along the x-axis, $S_x^*$, for a top-pumping geometry and the same value of the applied magnetic field. **d** Temperature dependence of $S_y^*$ for two different pumping geometries - top pumping (red) and substrate pumping (black) - and an applied magnetic field of 860 mT for the uncapped FeRh sample. In this case the normalisation factor used to extract $S_y^*$ was calculated using the electrical conductivity of the uncapped FeRh sample. Data in both (**c**) and (**d**) are normalised by the value of $S_y^*$(400 K) measured in FeRh-Pt in the top-pumping geometry. Data sets without the correction for THz outcoupling and information on the absorbed pump fluences are included in section S1 of the SI.

setup, where the $z$-axis is normal to the sample surface. Pump pulses propagate along the $z$-axis. The sample was mounted in an optical cryostat and the ambient temperature was varied in the range 20–420 K. We applied an external magnetic field along the $x$-axis using an electromagnet, with a field strength up to $|B| \approx 860$ mT. We pumped the sample from the substrate side (substrate pumping) or the capping layer side (top pumping). For the optical excitation, we used linearly-polarised 50 fs laser pulses with central wavelength of 800 nm.

The THz radiation emitted after optical pumping has mainly two origins: (i) the magneto-dipole component, which is directly related to the pump-induced change in magnetisation ($\frac{dM}{dt}$), and (ii) the electro-dipole component, which is related to the spin current generated, $j_s$, converted into a radiating sub-picosecond charge current via the ISHE. The detected far-field THz radiation, $S(\omega)$, is related to the THz field directly behind the sample, $E_{\text{THz-out}}$, by the convolution with the transfer function of the detection unit[17]. In the case of a purely electro-dipole emission[13]:

$$S(\omega) \propto E_{\text{THz-out}}(\omega) \propto C(\omega)E_{\text{THz-in}}(\omega) \propto A\theta_{\text{SH}}\lambda_s C(\omega)j_s(\omega). \quad (1)$$

Here, $E_{\text{THz-in}}(\omega)$ represents the THz electric field generated within the sample, $A$ is the optical fluence absorbed by the metallic stack at the pump wavelength of 800 nm, $\theta_{\text{SH}}$ is the spin Hall angle and $\lambda_s$ is the spin relaxation length. The parameter $C(\omega)$ describes the outcoupling of the THz field from the metallic stack:

$$C(\omega) \propto \frac{1}{1 + n_{\text{MgO}} + Z_0 \int_0^d dz\, \sigma(z)}. \quad (2)$$

where $n_{\text{MgO}}$ is the refractive index of the MgO substrate, $Z_0$ is the vacuum impedance, $d$ is the metal stack thickness, and $\sigma(z)$ is the conductivity distribution of the metal stack.

## Temperature dependence

In Fig. 1b we plot $S_y^*(T) = S_y(T)/[C(T)A(T)]$, the peak-to-peak amplitude of the emitted THz pulse transient normalised by the temperature-dependent THz outcoupling $C(T)$ and optical pump absorption $A(T)$, as described in detail in section S1 of the Supplementary Information (SI). The subscript $y$ indicates the pulse component polarised along the $y$-axis, perpendicular to the applied magnetic field. Normalising by $C(T)$ and $A(T)$ accounts for the temperature dependence of the sample's conductivity and refractive indices, therefore in the case of electro-dipole emission, the temperature dependence of $S_y^*$ essentially reflects the temperature dependence of the spin current, $j_s$.

Previous optical pump-THz emission studies on FeRh and FeRh-Pt[13,14,18], focused on the metamagnetic AF-FM phase transition near $T_{\text{(AF-FM)}}$ and only measured down to $T = 300$ K. The shaded area in Fig. 1b indicates the temperature region studied in those works. In this range, we observe the same behaviour as others, with the THz emission amplitude being almost fully suppressed below $T_{\text{(AF-FM)}}$ due to the reduction in volume of the FM phase. However, we extended the measurements by continuing to the low-temperature region, down to $T = 20$ K. This is deep within the AF phase and far below $T_{\text{(AF-FM)}}$ which is 370 K in our sample for the absorbed pump fluence of 2.4 mJ/cm². Surprisingly, as we decrease the temperature, $S_y^*(T)$ starts to increase again, above the value in the FM phase. This is despite the fact that there is only a tiny magnetisation in the AF phase, which we measured with SQUID (Superconducting Quantum Interference Device). Full data is shown in section S2 A of the SI.

Below $T_{\text{(AF-FM)}}$ the THz emission is linear with the pump fluence (see section S3 F. of the SI) and we see no threshold behaviour. These observations indicate that the increasing $S_y^*$ in the AF phase is not related to a heat-induced metamagnetic phase transition. From the specific heat of FeRh (0.35 J/g K)[19] we estimate that the transient

electron temperature rise upon optical pumping is ~130 K in the first 10 nm near the Pt interface at the pump fluence of 2.4 mJ/cm², too low to justify a phase transition at low temperature. Moreover, heat accumulation effects are minimal due to the low repetition rate of our laser (5 kHz), as also discussed in section S3 H. of the SI.

We confirm that the THz emission is of electro-dipole character, induced by a spin current, by measuring in both the top and substrate-pumping geometries. The results in Fig. 1c show that the THz emission has a dominant odd component when the sample is flipped with respect to the pump incidence side, in agreement with the symmetry of the ISHE. We see the ISHE symmetry both above and below $T_{\text{(AF-FM)}}$ which further confirms that in both the FM and AF phases the emission is due to spin currents. The smaller amplitude of the emission when pumping from the substrate side is because of the smaller absorbed pump energy in the region closer to the interface with Pt. On the same graph, we show that the THz emission is fully polarised along the $y$ direction, which is compatible with a polarisation of the spin currents parallel to the external field.

In Fig. 1d we show the same top- and substrate-pumping measurements performed on an uncapped FeRh sample without any Pt. The temperature dependence of the THz emission differs significantly from that in FeRh-Pt. In the ferromagnetic phase, the amplitude of the THz emission is more than a factor of six smaller. Moreover, it does not show the expected symmetry for electro-dipole emission; its polarity does not change for two flipped pumping geometries. This agrees with previous observations[12] and the signal is attributed to the magneto-dipole origin of the emission, which can become the main contribution in the absence of efficient spin-to-charge conversion via the ISHE. At $T < T_{\text{(AF-FM)}}$ the emission is still smaller but shows an increasing trend as the temperature is lowered. Quite remarkably, in this AF region, the flipped geometry changes sign as expected for spin currents converted via the ISHE. Bare FeRh does have an appreciable spin Hall angle and the quantity $\theta_{\text{SH}}\lambda_s$ has been measured to be twice as large in the AF phase than the FM phase[20]. Even so, this result indicates that in the AF phase where negligible spin current would normally be expected, there is actually a bulk contribution large enough to produce an electro-dipole emission. We note that the AF phase of FeRh usually has some residual FM order[21,22], often as a surface skin, but the contribution from these FM regions alone would be expected to be much smaller than the bulk FM phase.

## Magnetic field dependence

Figure 2a shows the magnetic field dependence of $S_y^*$ in the FeRh-Pt sample at fixed temperatures. In the ferromagnetic phase ($T = 400$ K), $S_y^*$ saturates with a small magnetic field, as we expect in a spintronic emitter picture where the pump-generated spin currents scale with the magnetisation of the ferromagnetic metal. In the antiferromagnetic phase ($T = 100$ K, $T = 200$ K) $S_y^*$ contains both a saturating behaviour and a linear increase. At a sufficiently high magnetic field we can describe the signal with $S_y^* = S_{y(\text{B-sat})}^* + S_{y(\text{B-linear})}^* B$. In Fig. 2b we plot $S_{y(\text{B-sat})}^*$ and $S_{y(\text{B-linear})}^*$ as a function of temperature with an applied field of +860 mT. In the considered field range $S_{y(\text{B-sat})}^*$ has a significantly larger amplitude with respect to $S_{y(\text{B-linear})}^*$ at low temperature. Also, differently from $S_{y(\text{B-sat})}^*$ that keeps increasing with decreasing temperature, $S_{y(\text{B-linear})}^*$ saturates at $T \approx 50$ K. The $S_{y(\text{B-linear})}^*$ component is consistent with the Zeeman splitting of AF states[23,24] allowing for non-zero net spin currents. The $S_{y(\text{B-sat})}^*$ component cannot be easily understood based on previous works, and in the following discussion we focus solely on this component.

The magnetic field dependence of $S_y^*$ suggests that residual ferromagnetism in the FeRh-Pt bilayer plays a role in the spin current generation below $T_{\text{(AF-FM)}}$. The presence of ferromagnetism at both FeRh interfaces, with the substrate and the capping layer, is well known[22,25–31]. Depth profile studies[21,22,25] have shown that these FM regions typically extend over a thickness of 3–5 nm and have a much

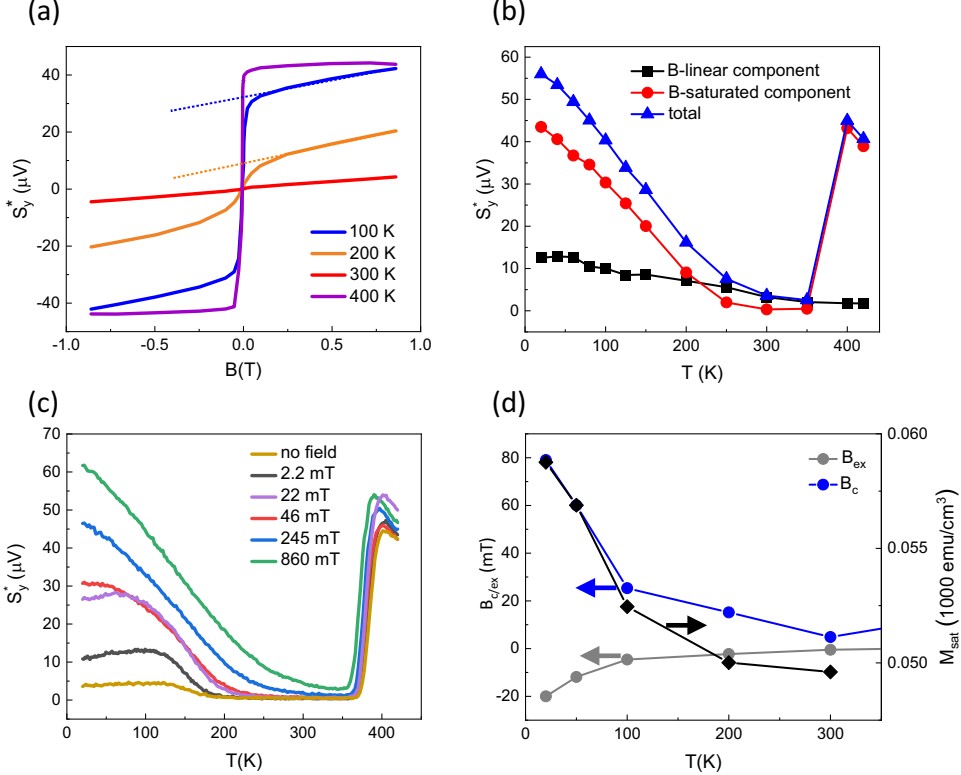

**Fig. 2 | Magnetic field dependence of the THz emission. a** THz emission amplitude measured as a function of in-plane magnetic field at selected temperatures. The dashed lines indicate the B-linear components of the signals. **b** Amplitude of both components $s_{y(B-sat)}$ and $s_{y(B-linear)}$ and their sum at +860 mT over the investigated temperature range. **c** Temperature dependence of $S_y^*$ for different magnetic fields up to +860 mT. The experiment was performed in the top-pumping geometry. Data sets without the correction for THz outcoupling and the information on the absorbed pump fluences are included in the SI. **d** Temperature dependence of $M_{sat}$, coercive field $B_c$ and exchange bias field $B_{ex}$ of the residual FM regions in FeRh-Pt.

lower average magnetic moment relative to the FM phase of FeRh. We confirm the presence of ferromagnetism in the low-temperature region in our samples using SQUID magnetometry (section S2 A. of the SI). In Fig. 2d we plot the saturation magnetisation $M_{sat}$ extracted from the SQUID data below $T_{(AF-FM)}$ and show that it increases slightly as we decrease the temperature below $T \approx 200$ K. The data shown in Fig. 2d is normalised by the total volume of FeRh, but even considering the normalisation by the effective volume of the ferromagnetic interfaces with a combined thickness of 10 nm, $M_{sat}$ in the AF phase remains significantly smaller than $M_{sat}$ for FM-FeRh, in agreement with previous studies[1,32–34]. We reproduce the temperature dependence of the magnetisation via atomistic spin dynamics simulations considering a monolayer of FM-FePd in contact with AF-FeRh, as discussed in more detail in section S2 B. of the SI. FePd alloying at the top interface is also confirmed by Scanning Transmission Electron Microscopy measurements in section S2 C. of the SI.

From the SQUID data we also extract a positive ($B_{c+}$) and negative ($B_{c-}$) coercive field value as a function of ambient temperature. Figure 2d shows that the coercivity, defined as $B_c = (B_{c+} - B_{c-})/2$, increases below 300 K, in line with other FM alloys[35–37]. At temperatures below 300 K we also observe a clear asymmetry between $B_{c+}$ and $B_{c-}$, indicated by a non-zero value of $B_{ex} = B_{c+} + B_{c-}$. This is likely due to an exchange interaction between the FM interface and the adjacent antiferromagnet.

In Fig. 2c we plot the temperature dependence of $S_y^*$ at different magnetic field values. Here we observe that while at the highest field values above $B_c$ (860 mT and 245 mT) the curves are described by the same temperature law, apart from a linear-in-field offset, at lower field values below $B_c$, $S_y^*$ is heavily suppressed, pointing again towards a role of the FM interface.

## Discussion

The temperature and magnetic field dependence of $S_y^*$ suggests that the residual magnetism, likely to exist at the interface with Pt, plays a key role in the enhanced spin emission at low temperature. However, the magnitude of $S_y^*$ in the AF phase cannot be explained by contributions from such a small magnetisation and small volume of the FeRh if only the small FM regions contribute to the spin currents. Therefore, we conclude that the antiferromagnetic phase must be contributing a significant spin current. For the antiferromagnet to act as a source of spin angular momentum time-reversal symmetry must be broken. The Zeeman splitting in the applied field provides this in a small way, and we have already identified this contribution as the linear term $S_{y(B-linear)}^*$. Optical pumping by itself does not break time-reversal symmetry. The magnetic difference frequency generation process[38,39] does not apply here because we pump along the (001) crystal axis which is not a three-fold rotation axis (FeRh belongs to the space group $Pm\bar{3}m$). The optical pumping therefore only results in the diffusion of spin-polarised hot carriers from the FM volume. In the standard spintronic emitter picture, only the transport of these carriers towards Pt is considered. We believe that transport towards the AF bulk of FeRh is instead important to consider in order to explain our results, as suggested by the proportionality of $S_y^*$ with the FeRh DC conductivity, $\sigma_{FeRh}$, below $T_{(AF-FM)}$ in Fig. 3a. In a spintronic emitter picture, instead, the temperature dependence of the spin current is dictated by the spin Hall physics of Pt and scales with its resistivity $\rho_{Pt}(T)$[40], please refer to section S3 I of the SI. Our observation suggests that charge transport in FeRh is important to the generation of the spin currents at low temperature. In Fig. 3b we plot the temperature dependence of $S_y^*/\sigma_{FePt}$. In the region of the phase transition, $S_y^*/\sigma_{FePt}$ scales with the drop in $M_{sat}$, as one would expect in a spintronic emitter picture. The significantly

(a)

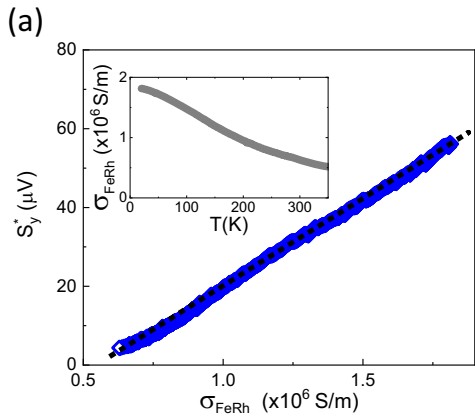

(b)

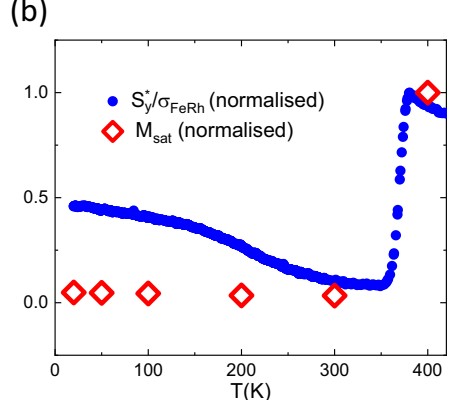

**Fig. 3 | Dependence of the THz emission on FeRh conductivity. a** $S_y^*$ measured in FeRh-Pt, pumping on the Pt side, as a function of FeRh conductivity, $\sigma_{FeRh}$ and for a value of the external field of 860 mT. The insert shows the temperature dependence of $\sigma_{FeRh}$[21]. **b** Temperature dependence of normalised $S_y^*/\sigma_{FePt}$ at an applied field of 860 mT (blue dots) and of normalised $M_{sat}$ (red diamonds).

higher rate of increase as the temperature is further lowered below $T_{(AF-FM)}$ reflects the different origin of the spin current in the AF phase.

We therefore suggest that the spin-polarised free carriers from the interfacial region have a strong polarising action on the FeRh lattice after optical pumping. Time-resolved photoelectron spectroscopy studies have shown that optical pumping of AF-FeRh results in a sub-picosecond alteration of the electronic structure that redistributes spin density from Fe towards Rh, producing a small transient Rh moment[41]. This is a manifestation of the optically-induced intersite spin transfer (OISTR) process[42–44]. Despite this redistribution of spin density, unless the system is macroscopically spin-biased, the transient Rh moments in the different unit cells will average to zero and will not result in any coherent action on the Fe spin-lattice.

The spin-polarised free carriers from the ferromagnetic interface provide this spin bias via s-d exchange, with their polarising action increasing with the spin lifetime within FeRh, as schematically shown in Fig. 4a. A recent work by Kang and co-workers[45] showed that the optically-induced AF-to-FM phase transition results in spin angular momentum absorption by the conduction electron bath of a Cu capping layer. In this work, we show instead that spin-polarised conduction electrons strongly destabilise the AF lattice and spin-polarise it also at temperatures much below the phase transition. To study the effect of a small, polarised, but transient moment being induced on the Rh sites, we performed atomistic spin dynamics simulations including a Landau Hamiltonian term which allows the magnitude of the spin moments to change and fluctuate[46–48]. We model the polarising effect as a Gaussian increase in the size of the Rh moments, in the spin-polarised direction, while the laser pulse is applied. A full description of the model is in sections S4 and S5 of the SI.

We perform spin dynamics simulation where we apply a weak perturbation that consists in a non-zero transient spin on the Rh site, less than $0.2\,\mu_B$ per Rh spin on average (Fig. 4b), (bottom right), and find a significant coherent spin pumping from the Fe spin-lattice (Fig. 4b), (top left). The sudden appearance of the Rh moment within the lattice breaks the time-reversal symmetry. The strong FeRh exchange amplifies the effect of the Rh and results in a net magnetisation of ~0.35 $\mu_B$ per Fe spin (Fig. 4b), (bottom left) which is much larger that the magnetisation induced by Zeeman splitting for the magnetic fields we apply. The Fe moments precess in the exchange field from the Rh, producing a spin pumping (Fig. 4b), (top row). To generate a net Fe moment, the exchange field must be greater than the spin flop field of the AF phase which we estimate to be 30 T. For an induced Rh moment of 0.2 $\mu_B$ the exchange field at the peak is around 200 T, although appearing for less than a picosecond (see section S5 of the SI for more details). We note that the longitudinal spin pumping

from both the Fe and Rh represents a negligible contribution (Fig. 4b), (middle row). In fact, the Rh moments contribute almost no direct spin pumping, differing from predictions at the phase transition[45], but their effect of breaking the time-reversal symmetry on the Fe spin-lattice is dramatic. This results in spin angular momentum being released from the antiferromagnetic phase, even without causing the metamagnetic phase transition. Increasing the applied magnetic field results in increased ordering of the ferromagnetic skin, until the magnetisation saturation point. The increasing magnetic order results in the higher polarisation of the initial spin current, leading to the increase in $S_y^*$, as we observe in our experiment. We simulate this by studying how the induced magnetisation (Fig. 4c) and spin pumping (Fig. 4d) increase with Δ, the amplitude of the transient Rh moment. We find that the induced magnetisation of both Fe and Rh are linear with Δ. For the spin pumping, we see that it is dominated by the Fe contribution and increases with Δ, in agreement with the experiments where larger magnetic fields (more strongly ordering the FM skin) and higher laser fluence both result in increases in the spin current. In the experiments, the Rh-mediated exchange-enhanced spin pumping saturates above the coercive field of the residual ferromagnetic interface (~80 mT), because of the spin polarisation of the currents generated by optical pumping saturates. The spin pumping that we measure exceeds that measured in ferromagnetic metal CoFeB, as shown in section S3 I of the SI. We predict that the amplitude of the spin pumping could be further enhanced by doping engineering to control the position and nature of the ferromagnetic regions such to maximise the anti-ferromagnetic volume involved in the spin emission. The possibility to generate high spin current pulses at low magnetic fields is important in the context of high-speed magnetic recording[49,50]. Achieving this with an antiferromagnet would suppress dipolar interactions, allowing growing the spin emitter in direct contact with the active bit-element.

## Methods
### Materials
The FeRh and FeRh-Pt samples were grown using DC magnetron sputtering on (001)-oriented MgO substrates. The substrates were annealed in situ overnight at 600 °C. After the annealing process, the FeRh thin film was deposited. The sample was then annealed in situ at 700 °C for 1 h. The samples were transferred to a different deposition chamber to deposit the Pt. In this chamber, the samples were reannealed at 700 °C for an hour and a half to refine the crystal structure and remove any defects on the surface that formed between the depositions. The sample was then left to cool to room temperature before the Pt layer was deposited.

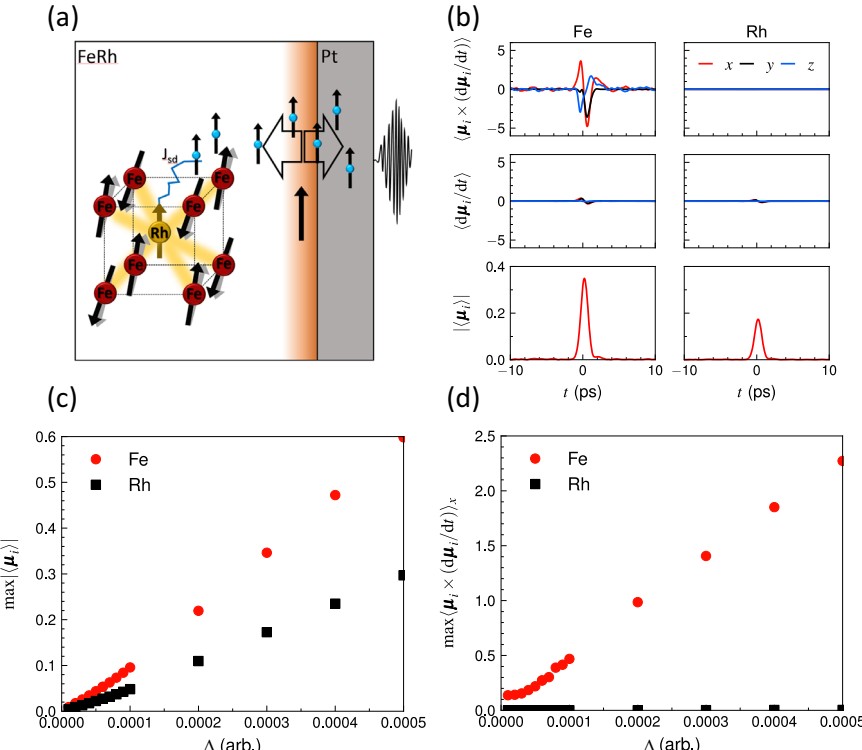

**Fig. 4 | Model of the enhanced spin pumping from antiferromagnetic FeRh.**
**a** Schematics of the mechanism that leads to spin pumping in the AF phase of FeRh. Optical pumping causes the diffusion of spin-polarised carriers from the ferromagnetic interface towards AF-FeRh, inducing a transient spin on the Rh atoms (shown by the brown arrow). The Rh spin acts with a torque on the Fe spins, shown by black arrows, and the magnetisation rises. The generated non-equilibrium spin in FeRh is pumped back into Pt and results in THz emission (not shown). **b** Dynamics of real (top) and imaginary (middle) parts of the spin pumping and the size of the induced magnetisation (bottom) in the Fe (left) and Rh (right) sublattices with $\Delta = 3 \times 10^{-5}$. **c** Maximum amplitude of the induced magnetisation in Fe (red circles) and Rh (black squares) as a function of the pulse amplitude ($\Delta$). **d** Size of the maximum in the real spin pumping along the $x$ (applied field) direction as a function of the pulse amplitude ($\Delta$).

## Experimental setup

800 nm femtosecond laser pulses with pulse width of 50 fs are generated from an amplified Ti:sapphire laser system with a repetition rate of 5 kHz. An electromagnet is used to apply an external field up to 0.85 Tesla. The resulting THz emission is detected using an electro-optic sampling technique with a 1 mm thick ZnTe crystal.

## Reporting summary

Further information on research design is available in the Nature Portfolio Reporting Summary linked to this article.

## Data availability

The datasets generated during and/or analysed during the current study will be made available in the Repository of the University of Cambridge Apollo at the address https://doi.org/10.17863/CAM.107667. The VAMPIRE software used for the four-spin model is available at https://vampire.york.ac.uk. The JAMS software used for the induced moment simulations is not currently open source while intellectual property issues are being resolved, but can be made available to individual researchers upon request.

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

## Acknowledgements

C.C. and J.B. acknowledge support from the Royal Society through University Research Fellowships. This project was supported by the Diamond Light Source and has received funding from the European Union's Horizon 2020 research and innovation programme under the Marie Skłodowska-Curie (grant agreement No. 861300) and the Engineering and Physical Sciences Research Council (grant numbers EP/V037935/1 and EP/X027074/1). Calculations were performed on ARC4, part of the High-Performance Computing facilities at the University of Leeds. CC thanks Dr. Samer Kurdi for the fruitful discussion. QR would like to thank Karel Výborný for the fruitful discussion on ref. 4 of the SI as well as for providing data necessary to get the interband conductivity (ref. 5 in the SI).

## Author contributions

Dominik Hamara: data curation, formal analysis, methodology, visualisation, writing-original draft. Mara Strungaru: formal analysis, software, visualisation, writing-original draft. Jamie R. Massey: methodology. Quentin Remy: formal analysis, software. Xin Chen: methodology, data curation. Guillermo Nava Antonio: formal analysis, methodology, writing review and editing. Obed Alves Santos: methodology. Michel Hehn: methodology. Richard F.L. Evans: supervision, visualisation. Roy W. Chantrell: formal analysis, software, supervision. Stéphane Mangin: formal analysis, methodology. Caterina Ducati: supervision, formal analysis. Christopher H. Marrows: methodology, supervision, funding acquisition. Joseph Barker: formal analysis, funding acquisition, software, writing-original draft. Chiara Ciccarelli: conceptualization, formal analysis, funding acquisition, project administration, supervision, writing-original draft.

## Competing interests

The authors declare no competing interests.
