## [Peer Review File · Nature Communications]

REVIEWER COMMENTS

Reviewer #1 (Remarks to the Author):

see attachment

Reviewer #2 (Remarks to the Author):

This paper reports terahertz (THz) emission spectroscopy is employed to investigate the picosecond spin pumping from metallic FeRh as a function of temperature. Combined the temperature, magnetic field dependent THz emission with atomistic spin dynamics simulations, this work demonstrated the antiferromagnetic spin-lattice is destabilized by the combined action of optical pumping and picosecond spin-biasing by the conduction electron population, which results in spin accumulation. For that reasons, I recommend the paper be published in Nature Communications after the authors have address the following questions/issues:

1. In “Optical Pump-THz emission from FeRh-Pt”part, Fig. 1 (a) is not the layout of the “THz pump experiment”.
2. The efficiency of THz field outcoupling $C(\omega, T)$, (SI, Eq. 4) can be described in the text, not only mentioned in the support information.
3. In Fig. 1 (c), what is the reason for non-symmetry of THz emission with top pumping and substrate pumping?
4. In Fig. 2 (c), please comment on the physical mechanism of the peak observed at around 390 K.
5. Please comment in the text, that how does the incident pump fluence affect the temperature of the sample.
6. In Fig.3 (a), why the THz emission has a linear correlation with the conductivity of FeRh, in addition, why not discuss the FeRh/Pt sample?
7. Please give a discussion on the laser induced antiferromagnetic-ferromagnetic phase transition of the FeRh after photo-excitation, which will affect the current experimental result or not.
8. In the paper, “A” represents optical pump absorption in Page 3 and represents the amplitude of the transient Rh moment. Please clarify it.
9. As the application of this work is THz emitter. Thus, the conclusion section maybe too simple, I would suggest the authors to outline some recent and important THz generation and applications with some related literatures, for example, PhotoniX 4, 28 (2023), PhotoniX 2, 12 (2021), Light: Advanced Manufacturing 2: 10(2021), Laser Photonics Rev. 2022, 16, 2100688, such that a more complete state of art of THz generations and applications can be provided for the reader.

Reviewer #3 (Remarks to the Author):

Review report for “Ultra-high spin emission from antiferromagnetic FeRh”

The authors of D. Hamara et al. present that a bilayer of an antiferromagnetic phase of FeRh and a heavy metal Pt can emit a strong THz radiation, which is an indication of spin current from FeRh to Pt, at low temperatures. This finding is surprising because the antiferromagnetic phase does not have a net magnetization. Previously, the spin-current-driven THz emission was reported with ferromagnets that have a strong net magnetization.

The authors propose an explanation of how the antiferromagnet FeRh can generate spin currents. However, their explanation is not convincing enough. Before publication of this work, I request a major revision in response to the following comments.

1. Because the THz emission shows a saturation behavior with the applied magnetic field, this observation is related to a small remaining ferromagnetic phase. But, the magnitude of the ferromagnetic phase, measured by SQUID, is too small to explain the large THz emission. To relate the small ferromagnetic phase and large THz emission, the authors argue that the antiferromagnet phase amplifies the spin current that was originally provided by the ferromagnetic phase. For the spin amplification, the authors propose three steps: first, a small ferromagnetic phase generates spin current to the antiferromagnetic phase; second, by the spin current, a transient moment is induced on the Rh site; third, by the exchange coupling between Rh and Fe, the Fe atomic moments precess around the Rh moments and produce spin current by spin pumping. However, one of the key findings of this work is the strong temperature dependence: the antiferromagnetic phase emits a strong THz only at low temperatures. I am wondering how the author's explanation of the spin amplification by the antiferromagnetic phase can explain the temperature dependence.

2. The authors argue that the Fe moment precesses around the Rh moment and generates spin current by spin pumping. However, in the antiferromagnetic phase, the Fe moment has two sublattices with opposite magnetization. Then, in my understanding, the spin currents from two sublattices are canceled out. Please explain how the net spin current can be generated from two sublattices of Fe.

3. From the magnetic-field dependence of Fig. 2, a small ferromagnetic phase that remains even at low temperatures has a key role in the THz emission. In addition, in the temperature above the transition temperature, 370~430 K, Fig. 1b, the THz emission from a strong ferromagnetic phase also has a strong temperature dependence. Therefore, I suspect that only the ferromagnetic phase could be responsible for the THz emission for the entire temperature range. For example, both conductivities of antiferromagnetic FeRh, ferromagnetic FeRh, and Pt will increase with decreasing temperature. Do the authors have an understanding of the temperature dependence in the ferromagnetic phase at 370~430 K?

4. The authors prepared a gradient structure of $\text{Fe}_{50}\text{Rh}_{46.8}\text{Pd}_{1.7}\text{Ir}_{1.5}(5)/\text{Fe}_{50}\text{Rh}_{47.1}\text{Pd}_{2.2}\text{Ir}_{0.7}(10)/\text{Fe}_{50}\text{Rh}_{47.2}\text{Pd}_{2.8}(15)/\text{Pt}(3.5)$. Is the gradient structure critical for the THz emission? If there is a significant change in the THz emission with different gradient structures, it would provide important evidence for the underlying mechanism.

5. For the fabrication of a bilayer, the authors grow FeRh on MgO substrate, then transfer it to another chamber to grow Pt on top of FeRh. Has the transfer been done in situ? Or does it experience exposure to air? Then, I have a concern about the interfacial oxidation issue.

The manuscript entitled "Ultra-high spin emission from antiferromagnetic FeRh" demonstrated by optical pump-THz emission spectroscopy that antiferromagnetic-FeRh is a spin current emitter. Two samples have investigated, i.e., MgO/FeRh(Pd,Ir) (30 nm)/Pt(3. nm) and MgO/FeRhPd (35 nm). The magnetic-field dependence of THz emission shows that there is a magnetic-field independent component and a component linearly depends on the magnetic-field. The magnetic-field linear part of spin current is generated due to the broken of spin-degeneracy by magnetic-field which has been studied before ([DOI: 10.1126/science.aaz42](https://doi.org/10.1126/science.aaz42), Nature 578, 70-74 (2020)). The focus of the present work is the magnetic-field independent part, which has a multi-steps mechanism as proposed by the authors theoretically: I) generation of the spin current in the residual ferromagnetic FeRh magnetization (maybe due to non-perfect growth?) by optical pumping, and II) action of spin polarization to antiferromagnetic FeRh by s-d exchange coupling. I summarise my comments/questions below:

- (1) The authored mentioned that '*the THz emission has a dominant odd symmetry when the sample is flipped*' as shown in Fig. 1c. However, looking into the data carefully, it only shows that the THz emission is only of opposite sign for top-pumping and substrate pumping geometries and $|S_y^*(\text{top-pumping})| > |S_y^*(\text{substrate pumping})|$ holds at all temperatures. However, this is not fully expected by ISHE, which should have $S_y^*(\text{top-pumping}) = -S_y^*(\text{substrate pumping})$. Why the signal amplitudes differ by almost 5 times for these two geometries? This is not properly clarified in the manuscript.
- (2) For the control sample, i.e., the pure FeRh sample, as shown in Fig. 1d, however, it seems that the symmetry $S_y^*(\text{top-pumping}) \sim -S_y^*(\text{substrate pumping})$ holds for temperature lower than the phase transition temperature. Why the signal for this sample is antisymmetric and expected by ISHE? Moreover, what is reason for the antisymmetric S_y^* signal in this sample? The authors mentioned that it could be due to the finite spin Hall angle of AFM-FeRh. Is this due to the spin pumping from residual ferromagnetic FeRh to the AFM-FeRh?
- (3) For the magnetic field dependent results shown in Fig. 3b, the authors compare the $S_y^*(\text{B-sat})$ and $S_y^*(\text{B-linear})$ in one figure for fixed B-field and have mentioned that $S_y^*(\text{B-sat})$ has a larger amplitude at all temperatures. In my opinion, it is not a fair comparison since the B-linear part depends on the magnetic-field one chooses.
- (4) Regarding proposed the theoretical model for $S_y^*(\text{B-sat})$, if it is true, can it also explain the T-dependence of the THz emission? i.e., an enhancement of spin current at low temperature. My expectation is probably that it couldn't, because the spins of AFM-FeRh becomes rigid due to the enhancement of anisotropy and exchange interaction. I will be more convinced if calculated results as a function of temperature could be shown. The other point is the residual ferromagnetic FeRh between Pt and AFM-FeRh (Fig. 4a). Can this layer (~3-5 nm according to the authors) block the excited spin current into Pt? Simply because the spin dephases fast within the range of spin diffusion length in the ferromagnetic layer due to exchange interaction and spin-orbit interaction.
- (5) The authors suggested that the principle of generating more efficient spin current can be achieved in specially crafted synthetic antiferromagnets. Could be the finite spin diffusion length in ferromagnetic in this case a problem? It would be great if the authors could explain the concept in more detail.

In summary, the observation of spin current emission of antiferromagnetic FeRh could be true. More efforts should be devoted to the understanding of the spin current generation.

Reviewer 1

The manuscript entitled "Ultra-high spin emission from antiferromagnetic FeRh" demonstrated by optical pump-THz emission spectroscopy that antiferromagnetic-FeRh is a spin current emitter. Two samples have investigated, i.e., MgO/FeRh(Pd,Ir) (30 nm)/Pt(3. nm) and MgO/FeRhPd (35 nm). The magnetic-field dependence of THz emission shows that there is a magnetic-field independent component and a component linearly depends on the magnetic-field. The magnetic-field linear part of spin current is generated due to the broken of spin-degeneracy by magnetic-field which has been studied before (DOI: 10.1126/science.aaz42, Nature 578, 70-74 (2020)). The focus of the present work is the magnetic-field independent part, which has a multi-steps mechanism as proposed by the authors theoretically: I) generation of the spin current in the residual ferromagnetic FeRh magnetization (maybe due to non-perfect growth?) by optical pumping, and II) action of spin polarization to antiferromagnetic FeRh by s-d exchange coupling.

We thank the Reviewer for her/his comments and questions, which helped improving the manuscript. We provide below a point-by-point reply, highlighting the changes we made in the article's text. Here, we just want to point out that there are many reports on residual ferromagnetism [Phys. Rev. Materials 4, 034410 (2020), Appl. Surf. Science 607, 154870 (2023), Phys. Rev. Materials 3, 124407 (2019), Scientific Reports 6, 22383 (2016), Appl. Phys. Lett. 100, 262401 (2012), Phys. Rev. B 82, 184418 (2010), Appl. Phys. 103, 07B515 (2008)] in epitaxial FeRh, both at the interface with substrate and capping layer. The stabilization of a ferromagnetic phase at the interface seems very hard to avoid even in high quality films and is explained as resulting from a combination of strain, chemical unbalancing and symmetry breaking [Scientific Reports 6, 22383 (2016)]. Pd doping also results in residual magnetism [Journal of Crystal Growth 438, 19 (2016)]. In our SQUID measurements we cannot resolve the different contributions to the magnetisation from the top and bottom interfaces. However, we observe that very similar magnetisation values as a function of temperature are obtained for the bare FeRh sample and FeRh sample capped with Pt, so we come to the conclusion that residual magnetism is largely intrinsic to FeRh and is not for example caused by alloying or interdiffusion with the Pt capping layer. In the new version of the Supplementary Information (SI) we have added a new section (S2-C), where we report on Scanning Transmission Electron Microscopy (STEM) measurements on a 35 nm thick FeRh sample with a nominally uniform Pd doping concentration of 2.8% across its thickness. These measurements show an increase in the Pd:Fe ratio in comparison to the Rh:Pd and Rh:Fe ratios at the top interface.

I summarise my comments/questions below:

(1) The authored mentioned that 'the THz emission has a dominant odd symmetry when the sample is flipped' as shown in Fig. 1c. However, looking into the data carefully, it only shows that the THz emission is only of opposite sign for top-pumping and substrate pumping geometries and $|S^*y(\text{top-pumping})| > |S^*y(\text{substrate pumping})|$ holds at all temperatures. However, this is not fully expected by ISHE, which should have $S^*y(\text{top-pumping}) = -S^*y(\text{substrate pumping})$. Why the signal amplitudes differ by almost 5 times for these two geometries? This is not properly clarified in the manuscript.

The smaller amplitude of the THz emission when pumping from the substrate is due to the smaller spin current injected into Pt in this pumping geometry. In the model we are proposing, antiferromagnetic spin pumping is triggered by optically-induced intersite spin transfer within the FeRh unit cell [Nature Comm. 12, 5088 (2021)] combined with s-d exchange with the spin polarised hot electrons diffusing from the ferromagnetic regions. If the spin diffusion length of hot carriers and antiferromagnetic magnons is smaller than the FeRh thickness, the interface with the MgO substrate,

although ferromagnetic, gives a negligible contribution to the spin pumping because the spin currents from this region will be heavily attenuated when reaching Pt (spin-to-charge conversion within antiferromagnetic FeRh is much less efficient (Fig. S5 (b))). On the other hand, because the thickness of FeRh is higher than the optical penetration length, when pumping from the substrate side, much less pump energy is deposited in the region closer to Pt, resulting in a smaller spin current contribution.

In summary, the largest contribution to the spin pumping comes from the region closer to Pt, with a thickness comparable to the spin diffusion length. In FeRh the proportionality of the spin pumping on FeRh conductivity (Fig. 3(a) of main article) suggests that the relevant spin diffusion length is that of hot carriers rather than magnons.

The absorbed pump energy in this region determines the amplitude of the spin currents. In the transfer matrix simulations below (section S1 of the SI for details) we plot the normalised pump absorption energy (nm^{-1}) within FeRh as a function of ambient temperature and for the two different pumping geometries. The colour scheme goes from red (higher absorbed energy per nm) to blue (lower absorbed energy per nm).

In these simulations we ignore conductivity gradients due to doping and interface effects but we can already see that when pumping from the substrate side, the interface with Pt is “cooler” than when pumping from the Pt side.

We point out that also in traditional ferromagnetic spintronic emitters the THz emission amplitude will be different for the two pumping geometries if the thickness of the ferromagnet is higher than the spin diffusion and optical penetration lengths. Below, we show THz emission measurements on CoFeB-Pt taken in our set-up (unpublished) for two different thicknesses of CoFeB. For a thickness of 3 nm (left graph), the THz emission amplitude is the same (with flipped sign) for the two pumping geometries because the whole thickness of CoFeB contributes to the spin currents that reach the Pt layer. On the other hand, for a thickness of 30 nm, the THz emission amplitude is reduced when pumping from the substrate. This is because the absorbed pump energy is largest near the substrate, and this is where the highest spin currents are generated, but these spin currents are attenuated when reaching the Pt. The difference between the two pumping geometries will be larger for smaller spin diffusion lengths or smaller optical penetration lengths (higher conductivity), as for FeRh.

In the main text we have added the sentence “The smaller amplitude of the emission when pumping from the substrate side is because of the smaller absorbed pump energy in the region closer to the interface with Pt.”

(2) For the control sample, i.e., the pure FeRh sample, as shown in Fig. 1d, however, it seems that the symmetry S^*y (top-pumping) $\sim -S^*y$ (substrate pumping) holds for temperature lower than the phase transition temperature. Why the signal for this sample is antisymmetric and expected by ISHE? Moreover, what is reason for the antisymmetric S^*y signal in this sample? The authors mentioned that it could be due to the finite spin Hall angle of AFM-FeRh. Is this due to the spin pumping from residual ferromagnetic FeRh to the AFM-FeRh?

We admit that we are unable to isolate a single effect behind the observed THz emission in uncapped-FeRh. The odd symmetry of the THz emission with respect to sample flipping is compatible with ISHE-induced electro-dipole emission. This is possible because FeRh has a non-negligible spin-Hall angle and it becomes the main ISHE element in the absence of Pt. In this case, residual magnetism both at the top interface and MgO interface potentially contribute to the spin-pumping since both interfaces are in contact with the main ISHE element (AFM-FeRh). At the interface with the MgO substrate we will also have a contribution from the paramagnetic spin Seebeck effect (also with odd-symmetry with respect to sample flipping). Moreover, we will have magneto-dipole contributions from the residual ferromagnetism (with even-symmetry with respect to sample flipping). Finally, symmetry breaking at the interfaces can lead to THz emission of electro-dipole type, as we have also recently observed in another antiferromagnetic metal [Communications Physics 6, 320 (2023)]. All these contributions are much smaller than the ISHE-induced electro-dipole emission in Pt but cannot be ignored when Pt is absent, making the analysis much more complex. On the other hand, when FeRh is capped with Pt we can interpret the THz emission with spin pumping from FeRh to Pt because the spin conversion efficiency of the Pt is much greater, making possible to ignore other contributions.

3) For the magnetic field dependent results shown in Fig. 3b, the authors compare the S^*y (B-sat) and S^*y (B-linear) in one figure for fixed B-field and have mentioned that S^*y (B-sat) has a larger amplitude at all temperatures. In my opinion, it is not a fair comparison since the B-linear part depends on the magnetic-field one chooses.

The Reviewer is completely right that the B-linear component will by definition grow linearly with field and eventually become larger than the B-sat one. The main message of this work is, however, that high spin pumping from the antiferromagnet is achieved at low applied fields above the coercive field of the ferromagnetic interface, 0.02 T at 100 K and 0.08 T at 20 K. The nature of the microscopic mechanism that leads to spin pumping in FeRh is very different to what discussed in Nature Communications 14, 538 (2023), where the linear Zeeman splitting of the magnon bands is necessary

for a net spin-pumping. Here, the antiferromagnetic lattice is polarised by much stronger s-d exchange interactions mediated by the Rh (please, refer to point (4)).

We also corrected the text specifying that “In the considered field range S^*y (B-sat) has a significantly larger amplitude with respect to S^*y (B-linear) at low temperature”.

(4) Regarding proposed the theoretical model for S^*y (B-sat), if it is true, can it also explain the T-dependence of the THz emission? i.e., an enhancement of spin current at low temperature. My expectation is probably that it couldn't, because the spins of AFM-FeRh become rigid due to the enhancement of anisotropy and exchange interaction. I will be more convinced if calculated results as a function of temperature could be shown.

We are grateful to the Reviewer for this question, the temperature dependence of the THz emission is a very important point that allows us to gain insight into the origin of the spin pumping. To understand how optically-induced spin pumping depends on temperature we define S^* in the main article, where effects connected to the temperature dependence of the optical absorption and THz outcoupling are renormalized. We find that the temperature dependence of antiferromagnetic spin pumping is determined by (1) the temperature dependence of the uncompensated magnetisation M_{sat} , as shown in Fig. 2d of the main text, and (2) the temperature dependence of the FeRh conductivity σ , as shown in Fig. 3a of the main article. The THz emission is suppressed if we apply a field below the coercive field (Fig. 2c of main text), which is temperature dependent (Fig. 2d of main text). The dependence on conductivity points towards the importance of bulk charge transport. Higher spin diffusion lengths for the hot carriers from the ferromagnetic regions mean that a larger volume of antiferromagnetic FeRh is spin-polarised through Rh-mediated s-d exchange and contributes to spin pumping.

As the Reviewer points out, the temperature dependence of the spin susceptibility of Rh and the Fe-antiferromagnetic lattice should be considered. Both the Rh spin susceptibility and the Fe lattice spin susceptibility will peak near the antiferromagnetic-ferromagnetic phase transition but will have a small temperature dependence in the temperature region we consider in our measurements. The Curie temperature of FeRh is ~ 700 K and we don't expect strong variations of the exchange constants of the antiferromagnetic phase in this region [PRB 83, 174408 (2011), Phys. Rev. Lett. 93, 197403 (2004)].

The rigid nature of the spins does not itself hamper the effect we are showing. This is because the sudden appearance of the Rh moment couples to the Fe through an exchange interaction which is also strong. Essentially, we are overcoming the critical field for a spin-flop transition in the AFM. The field exists for such a short time that the spin-flop state is only transient, but a net magnetisation is produced (Fig.4b main text – there is a net Fe magnetisation) in the AFM and a corresponding spin pumping. We can put this on a more quantitative footing. The FeRh AFM state is dominated by the antiferromagnetic J_5 interaction, and the effective exchange field felt by the Fe sublattices has a field strength:

$$H_E = \frac{8|J_5|}{\mu_{\text{Fe}}}.$$

The anisotropy field strength is:

$$H_A = \frac{2k}{\mu_{\text{Fe}}}.$$

The lower critical field for spin-flop state for a two sublattice antiferromagnet with easy axis (at T=0) is [Gurevich and Melkov, Magnetization Oscillations and Waves, CRC Press, 1996]:

$$H_0 > \sqrt{H_A(2H_E - H_A)}$$

where H_0 is a field strength along the easy axis. For our model this critical field is 30 T. Our applied field is much lower than this, meaning that the two AFM modes can be split only a small amount, leading to only a small spin current, proportional to the applied field as seen in normal AFMs. However, the spontaneously generated Rh moment causes the sudden appearance of an additional exchange field inside of the AFM. This field is not staggered (alternating between sublattices) but directional, due to the spin polarised current. Hence it acts on the antiferromagnetic Fe sublattices in a similar way to an applied field, but is much stronger. The field strength is approximately,

$$H_{Rh} = \frac{8|J_1|}{\mu_{Fe}} S_{Rh}$$

Our results show that we only generate $S_{Rh} \approx 0.2 \mu_B / 1.0 \mu_B$ (Fig. 4b main text), but nevertheless the peak Fe-Rh exchange field is therefore $H_{Rh} = 235$ Tesla, much greater than the spin-flop field. This explains why the Fe develops a net magnetisation, it is trying to spin-flop, although has nowhere near enough time to transition in the ~ 1 ps the field appears for. This also explains why the spin pumping is far larger than can be achieved from an applied field below the spin-flop field of an antiferromagnet, and more similar to the large change seen in spin pumping once the field is above the spin flop transition [Phys. Rev. Lett. 116, 097204 (2016)]. This very large field will also dwarf any small temperature dependence of H_E and H_A and so the temperature dependence observed must arise from a different part of the mechanism.

In summary the temperature dependence must come from both the electronic conductivity and the temperature dependence of the ferromagnetic regions which polarize the spin current. In the new version of the SI we have added section S5, where we estimate the exchange fields.

The other point is the residual ferromagnetic FeRh between Pt and AFM-FeRh (Fig. 4a). Can this layer (~3-5 nm according to the authors) block the excited spin current into Pt? Simply because the spin dephases fast within the range of spin diffusion length in the ferromagnetic layer due to exchange interaction and spin-orbit interaction.

We agree with the Reviewer that the interface quality and composition will affect spin pumping but we don't think it will block it completely. Residual magnetism will be partially quenched by optical pumping, reducing the effect of the exchange interaction in the time frame in which we measure spin pumping (~ 1 ps). Also, residual ferromagnetism is unlikely to form a continuous film at the interface [Phys. Rev. Materials 3, 124407 (2019)]. Instead, ferromagnetic patches will be separated by antiferromagnetic FeRh. These interstitial antiferromagnetic regions directly in contact with Pt will also contribute to spin pumping [Nature Communications 11, 275 (2020)].

(5) The authors suggested that the principle of generating more efficient spin current can be achieved in specially crafted synthetic antiferromagnets. Could be the finite spin diffusion length in ferromagnetic in this case a problem? It would be great if the authors could explain the concept in more detail.

What we meant here is that the ferromagnetic element in contact with FeRh, which is so crucial to trigger antiferromagnetic spin pumping, could be engineered to maximise the spin pumping. Currently, we are relying on residual magnetism that naturally forms at the interface with Pt, but we could imagine creating alternating layers of FeRh and ferromagnetic FePd or dispersed ferromagnetic

islands via doping engineering to guarantee that a larger volume of FeRh contributes to the spin pumping. We agree with the Reviewer that the term “synthetic” is misleading and we have therefore removed this sentence. We also point out that the large spin pumping we observe in FeRh is really connected to the nature of the Rh spin, which can be polarised by a combination of optical pumping and exchange field and which allows us to use the internal Rh-Fe exchange fields above 200 T (point 4 of this reply) to excite the antiferromagnetic modes. Injecting a comparable spin current into a different antiferromagnet would not result in the same values of the spin pumping.

In summary, the observation of spin current emission of antiferromagnetic FeRh could be true. More efforts should be devoted to the understanding of the spin current generation.

Reviewer 2

This paper reports terahertz (THz) emission spectroscopy is employed to investigate the picosecond spin pumping from metallic FeRh as a function of temperature. Combined the temperature, magnetic field dependent THz emission with atomistic spin dynamics simulations, this work demonstrated the antiferromagnetic spin-lattice is destabilized by the combined action of optical pumping and picosecond spin-biasing by the conduction electron population, which results in spin accumulation. For that reasons, I recommend the paper be published in Nature Communications after the authors have address the following questions/issues:

We thank the Reviewer for the positive recommendation and we provide below a point-by-point reply to her/his questions.

1. In “Optical Pump-THz emission from FeRh-Pt” part, Fig. 1 (a) is not the layout of the “THz pump experiment”.

We have changed the description into “The layout of the experiment”.

2. The efficiency of THz field outcoupling $C(\omega, T)$, (SI, Eq. 4) can be described in the text, not only mentioned in the support information.

We have followed the Reviewer’s advice and inserted the definition of $C(\omega, T)$ in the main text as well.

3. In Fig. 1 (c), what is the reason for non-symmetry of THz emission with top pumping and substrate pumping?

We kindly refer the Reviewer to our reply to point (1) by Reviewer 1 where we explain the reason.

4. In Fig. 2 (c), please comment on the physical mechanism of the peak observed at around 390 K.

We interpret the peak in correspondence of the phase transition with the combined contribution of ferromagnetic and exchange-enhanced antiferromagnetic spin pumping since in this region ferromagnetic and antiferromagnetic regions co-exist. The higher spin-susceptibility of the Fe and Rh atoms in this region will further amplify the antiferromagnetic contribution to spin-pumping. Finally, in the ferromagnetic region spin pumping will also scale with conductivity if charge spin transport is dominant over magnon spin transport, which explains the monotonic decrease of the spin pumping for increasing temperature above the phase transition. We also refer the Reviewer to our reply to point (3) from Reviewer 3 for a more detailed analysis.

5. Please comment in the text, that how does the incident pump fluence affect the temperature of the sample.

From the specific heat of FeRh (0.35 J/g K) [Phys. Rev. Mater. 5, 064412 (2021)] we estimate that the transient electron temperature rise upon optical pumping is ~ 130 K in the first 10 nm near the Pt interface at the pump fluence of 2.4 mJ/cm², too low to justify a phase transition at low temperature. We also estimate the increase in sample's temperature caused by accumulated heat to be below 10 K. This is discussed in section G of the SI where we compare the phase transition temperature measured in the THz emission experiments while cooling with that measured in THz transmission, where we are effectively doing a fast conductivity measurement without applying any pump.

We have added the following in the main text:

“ From the specific heat of FeRh (0.35 J/g K) [Phys. Rev. Mater. 5, 064412 (2021)] we estimate that the transient electron temperature rise upon optical pumping is ~ 130 K in the first 10 nm near the Pt interface at the pump fluence of 2.4 mJ/cm², too low to cause a phase transition at low temperature.”

6. In Fig.3 (a), why the THz emission has a linear correlation with the conductivity of FeRh, in addition, why not discuss the FeRh/Pt sample?

We apologise for the confusion, the THz emission we are considering here is actually that measured in the FeRh-Pt sample. We have now amplified the figure caption into “ S_y^* measured in FeRh-Pt, pumping on the Pt side, as a function of FeRh conductivity”. The proportionality of the spin pumping on FeRh conductivity is a central result that lead us to the development of the microscopic model, pointing towards the importance of bulk charge transport. Lower scattering results in higher spin diffusions lengths for the hot carriers from the ferromagnetic regions into the antiferromagnetic FeRh lattice. This means that a larger volume of antiferromagnetic FeRh will be spin-polarised though Rh-mediated s-d exchange and contribute to spin pumping.

7. Please give a discussion on the laser induced antiferromagnetic-ferromagnetic phase transition of the FeRh after photo-excitation, which will affect the current experimental result or not.

We ask the Reviewer to refer to our answer to point 5.

8. In the paper, “A” represents optical pump absorption in Page 3 and represents the amplitude of the transient Rh moment. Please clarify it.

We thank the referee for pointing this out. We have changed the symbol for the transient Rh moment to ‘ Δ ’.

9. As the application of this work is THz emitter. Thus, the conclusion section maybe too simple, I would suggest the authors to outline some recent and important THz generation and applications with some related literatures, for example, PhotoniX 4, 28 (2023), PhotoniX 2, 12 (2021), Light: Advanced Manufacturing 2: 10(2021), Laser Photonics Rev. 2022, 16, 2100688, such that a more complete state of art of THz generations and applications can be provided for the reader.

We agree with the Reviewer that the main manuscript was truncated too abruptly and we have added the following paragraph: “In the experiments, the Rh-mediated exchange-enhanced spin pumping saturates above the coercive field of the residual ferromagnetic interface (~ 80 mT), because the spin polarisation of the currents generated by optical pumping saturates. The spin pumping that we measure exceeds that measured in ferromagnetic metal CoFeB, as shown in section I of the SI. We predict that the amplitude of the spin pumping could be further enhanced by doping engineering to control the position and nature of the ferromagnetic regions such to maximise the antiferromagnetic volume involved in the spin emission. The possibility to generate high spin current pulses at low magnetic fields is important in the context of high-speed magnetic recording [51,52]. Achieving this

with an antiferromagnet would suppress dipolar interactions, allowing growing the spin emitter in direct contact with the active bit-element.”

Reviewer 3

Review report for “Ultra-high spin emission from antiferromagnetic FeRh”

The authors of D. Hamara et al. present that a bilayer of an antiferromagnetic phase of FeRh and a heavy metal Pt can emit a strong THz radiation, which is an indication of spin current from FeRh to Pt, at low temperatures. This finding is surprising because the antiferromagnetic phase does not have a net magnetization. Previously, the spin-current-driven THz emission was reported with ferromagnets that have a strong net magnetization. The authors propose an explanation of how the antiferromagnet FeRh can generate spin currents. However, their explanation is not convincing enough. Before publication of this work, I request a major revision in response to the following comments.

1. Because the THz emission shows a saturation behavior with the applied magnetic field, this observation is related to a small remaining ferromagnetic phase. But, the magnitude of the ferromagnetic phase, measured by SQUID, is too small to explain the large THz emission. To relate the small ferromagnetic phase and large THz emission, the authors argue that the antiferromagnet phase amplifies the spin current that was originally provided by the ferromagnetic phase. For the spin amplification, the authors propose three steps: first, a small ferromagnetic phase generates spin current to the antiferromagnetic phase; second, by the spin current, a transient moment is induced on the Rh site; third, by the exchange coupling between Rh and Fe, the Fe atomic moments precess around the Rh moments and produce spin current by spin pumping.

We thank the Reviewer for carefully checking the manuscript and we report a point-by point answer to her/his questions below.

However, one of the key findings of this work is the strong temperature dependence: the antiferromagnetic phase emits a strong THz only at low temperatures. I am wondering how the author’s explanation of the spin amplification by the antiferromagnetic phase can explain the temperature dependence.

We kindly refer the Reviewer to the answer to point (4) of Reviewer 1. In short, the excitation of antiferromagnetic modes occurs via s-d exchange mediated by the Rh spin and relies on a net spin polarisation in the conduction bands of FeRh. As detailed in the answer to point (4), the temperature dependence of the spin-pumping is determined by the bulk conductivity of FeRh σ , hence the spin diffusion length of hot carriers in the antiferromagnetic lattice and the uncompensated magnetisation M_{sat} .

2. The authors argue that the Fe moment precesses around the Rh moment and generates spin current by spin pumping. However, in the antiferromagnetic phase, the Fe moment has two sublattices with opposite magnetization. Then, in my understanding, the spin currents from two sublattices are canceled out. Please explain how the net spin current can be generated from two sublattices of Fe.

We suspect the Reviewer meant “precesses” rather than “presses”. The process is more similar to a spin-flop transition, as explained in details in our reply to point 4 from Reviewer 1. We kindly refer Reviewer 3 to this part.

3. From the magnetic-field dependence of Fig. 2, a small ferromagnetic phase that remains even at

low temperatures has a key role in the THz emission. In addition, in the temperature above the transition temperature, 370~430 K, Fig. 1b, the THz emission from a strong ferromagnetic phase also has a strong temperature dependence. Therefore, I suspect that only the ferromagnetic phase could be responsible for the THz emission for the entire temperature range. For example, both conductivities of antiferromagnetic FeRh, ferromagnetic FeRh, and Pt will increase with decreasing temperature. Do the authors have an understanding of the temperature dependence in the ferromagnetic phase at 370~430 K?

This is an important point that we carefully considered when developing our microscopic model. Higher conductivity means longer spin diffusion lengths and therefore a larger volume contributing to spin-pumping. This is true in the antiferromagnetic phase, where antiferromagnetic spin-pumping is triggered by the optically-induced diffusion of spin-carriers from the ferromagnetic regions. But, as the Reviewer points out, this is also true in the ferromagnetic phase if we consider that charge-mediated spin transport is dominant over magnon-mediated spin transport. In Fig. 3b of the main article, also shown below, we renormalize the contribution of the spin-diffusion length to the spin pumping by plotting S^*/σ and compare it with the temperature dependence of the magnetisation M_{sat} .

We make the following observations:

- Across the ferromagnetic-antiferromagnetic phase transition, S^*/σ and M_{sat} drop by the same factor in line with a standard spintronic emitter picture where the optically generated spin currents are proportional to the magnetisation, or more precisely to its time derivative dM_{sat}/dt . This is very different to what happens in the low temperature range, where S^*/σ grows by a significantly larger factor than M_{sat} , which we interpret as an important sign that a traditional spintronic-emitter picture cannot explain the enhanced spin pumping at low temperature. Instead, we explain our results with spin pumping from the antiferromagnetic regions. In this case spin pumping is started by the spin currents generated in the ferromagnetic regions, proportional to dM_{sat}/dt , but is exchange-enhanced via the Rh spin (please, refer to our answer to point (4) by Reviewer 1).
- In the ferromagnetic phase S^*/σ flattens out after a peak. We interpret the peak in correspondence of the phase transition with the combined contribution of ferromagnetic and exchange-enhanced antiferromagnetic spin pumping since in this region ferromagnetic and antiferromagnetic regions co-exist. The higher spin-susceptibility of the Fe and Rh atoms in this region will further amplify the antiferromagnetic contribution to spin-pumping.

Finally, we want to comment on the fact that the dependence of spin pumping on conductivity reflects the higher volume contributing to the generation of the spin-currents when spin-conductivity increases. However, even if the whole volume of FeRh was contributing to spin pumping we would never be able to justify the magnitude with a standard spintronic emitter picture because of the much smaller magnetisation. On the other hand, if we assume residual ferromagnetism to be confined in the first nm's in contact with the two interfaces, in order for the

residual magnetisation to be comparable with the magnetisation of ferromagnetic FeRh, we would need to assume that it extends for an overall depth (both interfaces combined) of ~ 1 nm. This is below the spin-diffusion length, which makes the dependence on conductivity meaningless if we consider these ferromagnetic regions to be the only source for the spin currents.

4. The authors prepared a gradient structure of $\text{Fe}_{50}\text{Rh}_{46.8}\text{Pd}_{1.7}\text{Ir}_{1.5}(5)/\text{Fe}_{50}\text{Rh}_{47.1}\text{Pd}_{2.2}\text{Ir}_{0.7}(10)/\text{Fe}_{50}\text{Rh}_{47.2}\text{Pd}_{2.8}(15)/\text{Pt}(3.5)$. Is the gradient structure critical for the THz emission? If there is a significant change in the THz emission with different gradient structures, it would provide important evidence for the underlying mechanism.

We have followed the Reviewer’s suggestion and repeated the measurements on a doped FeRh film with same thickness and doping gradient, but inverted. The new Scanning Transmission Electron Microscopy measurements in section S2 C of the SI align with our prediction that residual ferromagnetism comes from FePd alloying at the top interface. We therefore expect that decreasing the Pd doping concentration at this top interface would result in a drop in THz emission amplitude. The two samples measured have the following compositions:

FeRh35 (additional) - $\text{MgO}/\text{Fe}_{50}\text{Rh}_{46.8}\text{Pd}_{1.7}\text{Ir}_{1.5}(5)/\text{Fe}_{50}\text{Rh}_{47.1}\text{Pd}_{2.2}\text{Ir}_{0.7}(10)/\text{Fe}_{50}\text{Rh}_{47.2}\text{Pd}_{2.8}(15)/\text{Pt}(3.5)$

FeRh36 (main article)- $\text{MgO}/\text{Fe}_{50}\text{Rh}_{47.2}\text{Pd}_{2.8}(15)/\text{Fe}_{50}\text{Rh}_{47.1}\text{Pd}_{2.2}\text{Ir}_{0.7}(10)/\text{Fe}_{50}\text{Rh}_{46.8}\text{Pd}_{1.7}\text{Ir}_{1.5}(5)/\text{Pt}(3.5)$

The two samples have very similar bulk properties (crystal quality and conductivity, which determines the THz outcoupling), but differ in the interface regions. Below we compare the temperature dependence of the THz emission amplitude S and spin pumping S^* in the two samples, normalised to the values in the ferromagnetic phase. The first thing we notice is that in sample FeRh35, the phase transition is shifted to higher temperatures. This is in agreement with the fact that Ir-rich FeRh has a higher transition temperature than Pd-rich FeRh [APL Mater. 3, 041802 (2015)] and confirms that only the volume closer to the Pt interface is involved in the spin-pumping. Also, we see that as temperature is decreased below the transition temperature, we measure a lower value of the spin pumping in FeRh. This shows that the chemical composition of the interface with Pt is important. The fact that spin pumping decreases with Pd concentration provides further evidence that residual magnetism resides in FePd alloying. One note, when we calculate S^* from the THz outcoupling we consider the bulk conductivity measured by 4-point electrical methods, however the different gradient in doping in FeRh35 and FeRh36 will result in a different gradient in bulk conductivity, which might slightly affect the values of the calculated optical absorption and THz outcoupling in the two samples.

We have added a section in the SI, section S3 G.

5. For the fabrication of a bilayer, the authors grow FeRh on MgO substrate, then transfer it to another chamber to grow Pt on top of FeRh. Has the transfer been done in situ? Or does it experience exposure to air? Then, I have a concern about the interfacial oxidation issue.

The sample has been exposed to air during this process. However, FeRh, owing to its high Rh content, is extremely resistant to oxidation [Phys. Rev. Materials 4, 123402 (2020)]. The sample is reannealed after being transferred to the second chamber to restore chemical order and remove any native oxide that may have formed at the surface.

REVIEWERS' COMMENTS

Reviewer #1 (Remarks to the Author):

The authors have addressed my comments seriously. Glad to know that other reviewers have similar concerns as me. In this case, the work deserves publication. I am also looking forward to see more interesting work using antiferromagnets as spin current source.

Reviewer #2 (Remarks to the Author):

All my concerns are revised well. I recommend the paper be published in Nature Communications.

Reviewer #3 (Remarks to the Author):

The revised manuscript provided additional experiments and explanations to clarify their arguments. I think their explanations are still not convincing enough. However, the experimental finding (large spin current from the antiferromagnetic phase of FeRh at low temperature) is surprising because it cannot be explained by the conventional understanding. Considering the novelty of this surprising experiment, I recommend the publication of this work in Nature Communications.